# Lightweight individual cow identification based on Ghost combined with attention mechanism

Lili Fu[1‡], Shijun Li[2‡], Shuolin Kong[1], Ruiwen Ni[1], Haohong Pang[1], Yu Sun[1], Tianli Hu[1], Ye Mu[1], Ying Guo[1], He Gong[1] *

1 College of Information Technology, Jilin Agricultural University, Changchun, China, 2 College of Electronic and Information Engineering, Wuzhou University, Wuzhou, China

‡ LF and SL share first authorship and also contributed equally to this work.
* gonghe@jlau.edu.cn

**Data Availability Statement:** Our dataset has been uploaded to figshare, and the DOI is 10.6084/m9. figshare.16879780.

## Abstract

Individual cow identification is a prerequisite for intelligent dairy farming management, and is important for achieving accurate and informative dairy farming. Computer vision-based approaches are widely considered because of their non-contact and practical advantages. In this study, a method based on the combination of Ghost and attention mechanism is proposed to improve ReNet50 to achieve non-contact individual recognition of cows. In the model, coarse-grained features of cows are extracted using a large sensory field of cavity convolution, while reducing the number of model parameters to some extent. ResNet50 consists of two Bottlenecks with different structures, and a plug-and-play Ghost module is inserted between the two Bottlenecks to reduce the number of parameters and computation of the model using common linear operations without reducing the feature map. In addition, the convolutional block attention module (CBAM) is introduced after each stage of the model to help the model to give different weights to each part of the input and extract the more critical and important information. In our experiments, a total of 13 cows' side view images were collected to train the model, and the final recognition accuracy of the model was 98.58%, which was 4.8 percentage points better than the recognition accuracy of the original ResNet50, the number of model parameters was reduced by 24.85 times, and the model size was only 3.61 MB. In addition, to verify the validity of the model, it is compared with other networks and the results show that our model has good robustness. This research overcomes the shortcomings of traditional recognition methods that require human extraction of features, and provides theoretical references for further animal recognition.

## Introduction

As dairy farming continues to expand, it has led to dairy farm management monitoring becoming the most important issue. Traditional manual monitoring of livestock with high accuracy and consistency cannot be performed by experience and eyesight alone [1], and the

**Funding:** The author(s) received no specific funding for this work.

**Competing interests:** The authors have declared that no competing interests exist.

identification of cows and statistical cow data by manual means is completely unsuitable for modern, large-scale dairy farming [2]. Traditional methods that do not rely on manual identification of livestock are permanent damage methods with inscribed ear branding, external tagging methods with ear tags, and external device tagging methods represented by radio frequency identification (RFID) [3]. These methods can cause irreversible harm to livestock and have the disadvantage of being unsustainable and reusable [4–7]. As a result, there is a growing need to replace traditional recognition methods with computer vision techniques [8].

In recent years, due to the maturity of deep learning technology, intelligent individual recognition technology based on deep learning technology has been applied to various fields, and intelligent technology has been widely used, especially in the field of image recognition has achieved fruitful application results [9, 10]. The deep learning methods, represented by convolutional neural networks, can accurately learn, forecast and classify targets in images [11]. Bello et al. [12] proposed a method for detection and monitoring of cows using computer vision technology, capable of tracking and identifying cow targets in video experiments with 89% recognition accuracy. Hu et al. [13] used yolo to detect each part of the cow, and then segmented each part of the cow using a partial segmentation algorithm with frame difference and segmentation span analysis, trained different recognition models separately, and then performed feature fusion to finally identify the individual cow identity accurately, with a final recognition accuracy of 98.36%. Mcdonagh et al. [14] performed continuous video monitoring of 46 cows to obtain datasets of different behaviors, and used convolutional neural networks to recognize and classify their behaviors with a final average recognition accuracy of 86.42%. Li et al. [15] proposed a basic motion behavior based on cow skeleton and a hybrid convolutional algorithm that effectively controls the number and robustness of model parameters while increasing the depth of the 3D convolutional network. Experimentally, 300 cow videos containing three specific motion behaviors were selected for testing, and the results showed that the final classification ACC of their method was 91.80% after 5-fold cross-validation. Shen et al. [16] proposed a method for real-time monitoring of cow regurgitation behavior based on edge computing. The three-axis acceleration signals of cows are collected and processed in real time using an edge device designed by the authors, adaptive thresholds are determined, and finally, the real-time identification of cow regurgitation behavior is accomplished at the edge device side, and the method does not require a lot of computational time and resources. Li et al. [17] proposed a convolutional neural network-based method for automated and accurate individual recognition of cows, using a residual learning inverse convolutional network to obtain a training dataset after denoising the cow images, and an improved InceptionV3 network as the main network for training to recognize individual cows with the cow tail pattern as the recognition feature. Achour et al. [18] developed a non-invasive system based entirely on image analysis to identify individual cows and their behavior using the cow head pattern as a mention feature. However, the small area of the head heel has limitations as a feature point for identifying individual cows, which has an impact on the final recognition results. Xiao et al. [19] used a modified mask R-CNN to segment and extract the back pattern of cows for cows in natural barns, and then trained an SVM classifier using a dataset consisting of patterns, and finally identified individual cows.

In this research, a lightweight convolutional neural network model using a combination of Ghost and CBAM is constructed, and the model is trained using our own dataset that was taken in a real barn, which represents a challenging machine vision situation where cows are photographed under different lighting, pollution and complex backgrounds. In this paper, we demonstrate the effectiveness of our model on complex and variable data sets and provide a systematic analysis of the various modules of the model.

## Materials and methods

### Image acquisition and expansion

The experimental data of this research were collected from Dongfeng cattle farm in Jilin Province and Changchun Boyu agricultural cattle breeding training base. The acquisition took place in July 2021 with a Canon EOS 5D Mark II and a maximum resolution of 5616×3744 pixels. The images of Holstein cows were taken from different angles in a real cattle environment with a handheld camera while the cows were standing and moving freely. Holstein cows have obvious black-and-white patterns, and the area with the most black-and-white interlacing is the cow's side, which can facilitate the recognition of convolutional neural network compared with other parts. In addition, a large amount of data is needed to train the convolutional neural network, and a larger area of black-and-white patterns on the side of the cow taken from different angles can also provide more data for training the convolutional neural network. A total of 13 cows were collected in this study, containing 3772 images of cows with complex backgrounds, and the cow numbers were customized as 1 to 13, as shown in Fig 1.

Since the size of the dataset has a great impact on the performance of the training network, the model is prone to overfitting when the feature space dimension of the samples is larger than the number of training samples [20]. To enhance the robustness and generalization of the network, a python scripting algorithm was used to divide the dataset into a 70% training set, a 20% validation set, and a 10% test set, the python scripting algorithm automatically divides datasets quickly and efficiently without overlapping images between datasets. Usually, model training requires sufficient sample size to avoid overfitting of the model [21], therefore, data enhancement is performed on the training set, and the traditional ways to expand the data set are image flipping, random cropping, and color dithering, etc. In this study, image rotation is chosen to randomly flip the data from three angles respectively, and the final sample size of the training set is increased to three times of the original one.

### Methods

In this research, ResNet50 is selected as the skeleton network, and firstly, ResNet50 is reduced so that the number of clump depth layers of the model is reduced, and then a series of improvements of the model are carried out on this basis. Considering cows with distinct and large range of feature points, features are extracted using null convolution in the first layer of the network. Then a plug-and-play ghost module is introduced into the model to make the model lightweight. Since the twisting during the cow's movement causes a certain degree of distortion of the spots, channel space attention is used in the model so that the model pays attention not only to the channel information of the features but also to the spatial information

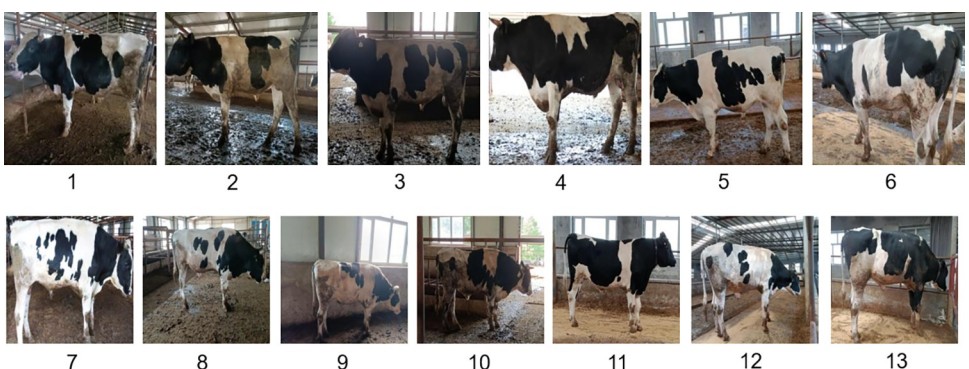

**Fig 1. Individual data from 13 cows used in this research.**

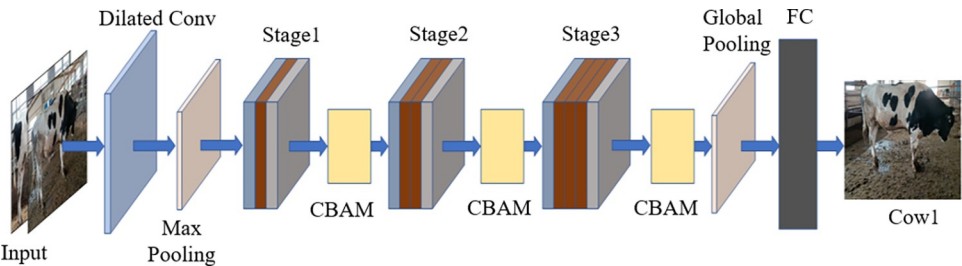

**Fig 2. The model structure built in this research.**

of the features. The final part of the model uses an optimization structure of a global pooling layer plus a fully connected layer to control the computational load of the model, and the model structure is shown in Fig 2. Stage1, Stage2 and Stage3 are the main structure of the model, and different numbers of ghosts are added in an incremental manner, following the rule that the further the convolutional neural network is in the forward propagation process, the more feature maps are added and the more convolutional kernels are required, which greatly reduces the number of model parameters while ensuring the depth of the model.

## Dilated convolution

In the original Resnet50 model, the first layer uses a 7×7 ordinary convolution to extract features, however, the large size of the convolution will weigh down the number of parameters of the model. To control the number of model parameters and to keep the perceptual field of the output unit constant, we choose to use dilated convolution instead of normal convolution. In the model, a 3×3 ordinary convolution is used, and its dilation rate is set to 3, which increases the receptive field of the convolution kernel, so that the 3×3 ordinary convolution becomes a 7×7 dilated convolution with an effective receptive field of 7×7, as shown in Fig 3, and the number of parameters of the model does not increase, but still only the computational effort consumed by the 3×3 convolution, which can effectively reduce the number of parameters of the model while preserving the large receptive field.

## Ghost module

The stacked convolutional blocks will extract a large number of feature maps with many duplicate features resulting in feature redundancy and a large number of model parameters. In our experiments, considering that cows have obvious speckle characteristics, we discard Bottleneck2 inside each stage of the model and choose to insert the plug-and-play GhostBottleneck

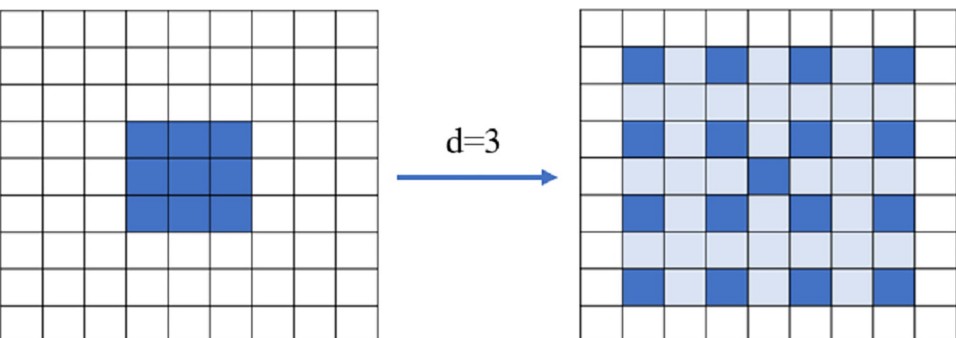

**Fig 3. Dilated convolution of 7×7 size used in the model.**

between Bottleneck1 and Bottleneck2 in each stage to reduce the number of parameters and computation of the model. This module was proposed by Han et al. [22], using the linear variation in the module to generate feature maps at a small cost can effectively solve the problem of redundant feature maps in the model. Ghost uses a small number of convolution kernels to extract features from the Bottleneck1 output, and then further performs a cheaper linear variation operation on this part of the feature map, and finally generates the final feature map as the input to Bottleneck2 by the concat operation, as shown in Fig 4.

## Convolutional block attention module

We choose to introduce improved CBAM modules after each stage of the model to improve the recognition accuracy of the model, and a total of three CBAM modules are introduced. The Convolutional block attention module (CBAM) attention mechanism proposed in 2018 takes into account the features of the model channel and space, allowing the network to focus on the "what" of the image as well as the "where" of the objects in the image [23]. The CBAM takes into account the features of the model channel and space, and the use of this attention in individual cow identification allows the model to better focus on each cow's spot and find where the spot is located, with the structure shown in Fig 5. The CBAM module introduced in the model can help the model assign different weights to each part of the input to extract more critical and important information, so that the model can make more accurate judgments without bringing more overhead to the model's computation and storage. After CBAM, the new feature map will get the attention weights in the channel and spatial dimensions, which greatly improves the connection of each feature in the channel and space, and is more conducive to extracting the effective features of the target.

# Results and discussion

## Experimental environment and parameter settings

The deep learning framework used in this experiment was pytorch 1.8.0. The version of Torchvision was 2.2.4 and the computer configuration was an Intel Core i7-8700 CPU running at

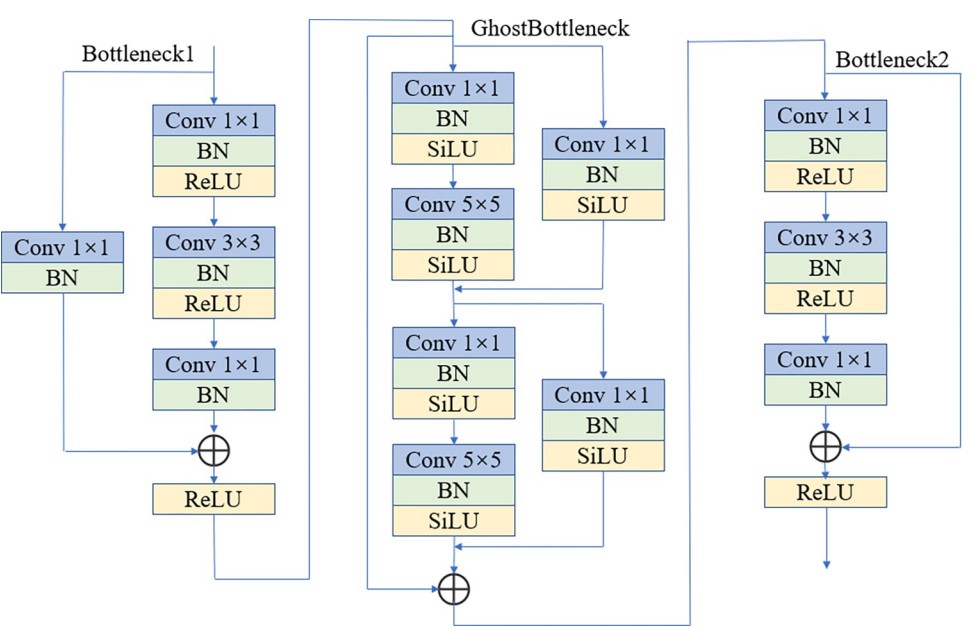

**Fig 4. The GhostBottleneck structure used in the model.**

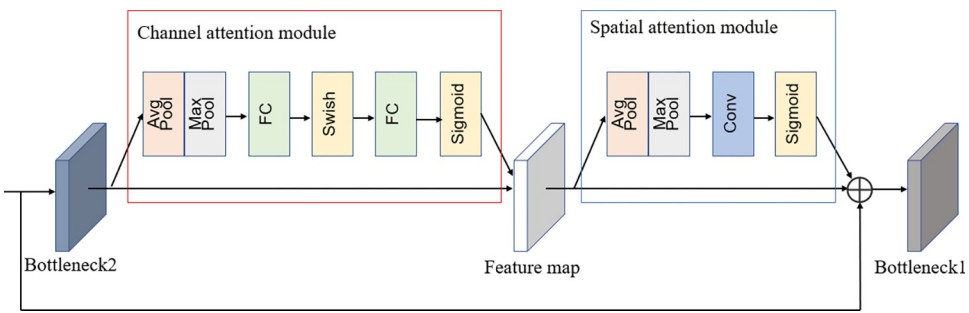

**Fig 5. The CBAM structure used in the model.**

3.20 GHz. It was equipped with an Nvidia GeForce GTX 1080 graphics card. The adapter was an Intel UHD Graphics 630. The software was CUDA API version 0.9.10, based on the Python 3.8.3 programming language and integrated with the PyCharm2020development environment.

To better evaluate the difference between the true and predicted values, a batch training approach was used to divide the training and testing process into multiple batches, each containing 32 images. The loss function uses cross-entropy loss, and the weight initialization method uses Xavier with an initialization bias of 0. The hyperparameters are set as shown in Table 1, where the initial learning rate of the model is 0.001 and the learning rate decreases by 0.1 every 10 epochs. the model uses a softmax classifier, and the input images are uniformly compressed to 224 × 224 pixels using an interpolation algorithm in order to obtain a high training speed while maintaining a good classification rate.

## Ablation experiments

Ablation experiments were conducted on this experimental model, and the same data set and experimental environment were used to train the model in each stage separately to prove the effectiveness of the used modules, and the results are shown in Table 2. At the beginning of this experiment, ResNet50 was layer-reduced to obtain Model A. The number of parameters was greatly reduced, and the model recognition accuracy was not affected, the improvement and construction of the model based on the A. Model B is to use the first layer of the model using dilated convolution instead of normal convolution to extract features using large sensory fields to improve recognition accuracy to a certain extent, while the number of model parameters is reduced. Model C uses the plug-and-play Ghost module on top of B. Using the linear operation of Ghost, the number of parameters of the model is reduced to 923,125 without reducing the number of feature maps, and the model size is significantly reduced. Finally, CBAM attention mechanism is introduced after each stage to ensure that the number of model

**Table 1. Hyperparameter setting.**

| Hyperparameters | Values |
|---|---|
| Classes | 13 |
| Batch size | 32 |
| Epoch | 50 |
| Optimizer | SGD |
| Learning rate | 0.001 |
| Momentum | 0.9 |

**Table 2. Add module comparison results.**

| Model | Dilated Convolution | Ghost | CBAM | Accuracy (%) | parameters | Model size (MB) |
|-------|---------------------|-------|------|--------------|------------|-----------------|
| ResNet50 | | | | 94.33 | 23,534,669 | 89.78 |
| A | | | | 94.88 | 1,694,925 | 6.47 |
| B | √ | | | 95.51 | 1,687,437 | 6.44 |
| C | √ | √ | | 96.54 | 923,125 | 3.52 |
| Ours | √ | √ | √ | 98.58 | 947,226 | 3.61 |

parameters grows within our acceptable range and to improve the model recognition accuracy, our model final recognition accuracy is 98.58%, which is 4.25 percentage points higher than the original ResNet50, the number of model parameters is reduced by 24.85 times, and the model size is only 3.61 MB. This result proves the effectiveness of the module used in our model.

## Model performance validation

To verify the effectiveness of the model, we compare it with other network models respectively, keeping the above training parameters and training methods, and train the convolutional neural network model using a cow dataset in a complex context, and the results are shown in Table 3. In the classical CNN model, ResNeXt and Google net have good results in recognition accuracy, but are still lower than the recognition accuracy of our model. Our model is much lower than ResNeXt and GoogLeNet in terms of both FLOPs and model size. While Shuffle-NetV2 and MobileNetV3, which are famous for their lightweight, do have lower FLOPs than our model, but the single use of the size of FLOPs is not an objective measure of whether the model is lightweight or not [24]. From the model size, our model size is only 3.61MB and the recognition accuracy is much higher than ShuffleNetV2 and MobileNetV3. The reason why MobileNetV3 has a lower recognition accuracy is due to the use of NAS technology in Mobile-NetV3 model to automatically optimize the model with ImageNet as the dataset [25], which will greatly reduce the number of parameters of the model, but the generalization of the model is not high, and therefore it does not show a high recognition accuracy on cow dataset. In comparison with newer models, two models proposed in 2021, EfficientNet v2 and PVT, are worse than ours in terms of both recognition accuracy and FLOPs. PVT, as the backbone network of Transformer, has almost no bias induction, leading to higher overfitting risk on small data as well as high computational effort [26], with a computational speed of 1.86G and a model size of 23.98MB, which is not light enough. As can be seen in Fig 6, although GoogLeNet and our model are almost equal, GoogLeNet has instability in validation. Our model achieves 70% in the first iteration with better performance. It can be seen that the lightweight convolutional neural network model constructed in this study can greatly reduce the number of model

**Table 3. Performance analysis results of different models.**

| Architecture | Validation accuracy (%) | FLOPs | Model size (MB) | Time (s) |
|--------------|-------------------------|-------|-----------------|----------|
| ResNeXt | 95.65 | 4.26G | 87.76 | 0.2703 |
| GoogLeNet | 97.96 | 2G | 21.04 | 0.2088 |
| ShuffleNetV2 | 85.87 | 591.08M | 5.26 | 0.3068 |
| MobileNetV3 | 83.82 | 262.12M | 8.52 | 0.2270 |
| EfficientNet v2 | 87.64 | 2.975G | 22.39 | 0.4006 |
| PVT | 90.57 | 1.86G | 23.98 | 0.2070 |
| Ours | 98.58 | 627.69M | 3.61 | 0.2410 |

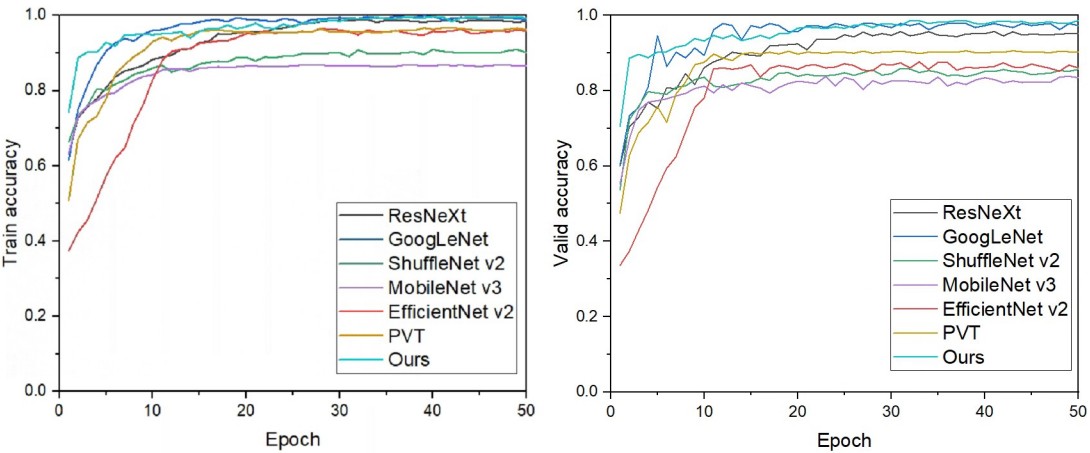

**Fig 6. Comparison results of different network models: A is training accuracy of model, B is validation accuracy of model.**

parameters while ensuring recognition accuracy, which can identify individual cows faster, save recognition time and breeding costs in real dairy farms, and better meet the application needs of real complex dairy farm environments.

## Comparison with lightweight models used by other researchers

The method of identifying cows in this study was compared with the methods proposed by other researchers studying cows and the results are shown in Table 4. The literature [27] achieved recognition of cow rump by fine-tuning mobilenet, and the final recognition accuracy was 99.76%, but the mod size was 9.25 MB, which was more than two times larger than our model. The literature [28] uses ReXNet 3D for cow behavior recognition and its model size is 14.3 MB with low accuracy, which is 10.69 MB larger than our model, in addition, its FLOPs are 15.8 G, which is also much larger than our model. Finally, compared with our previous work [29], The model in this study has a good performance improvement in both recognition accuracy and model lightweight, overcoming the deficiencies in recognition accuracy caused by complex backgrounds and speckle distortion from the spatial dimension, improving 0.63 percentage points compared to previous work, and also reducing the model size by 4.97 MB using the linear operation of ghost, making the model more lightweight.

## Conclusions

In order to further improve the accuracy, stability and real time of cow identification, and to explore new methods of cow identification with stronger practical application capabilities to promote the development of intelligent cattle farming methods. In this research, we propose a convolutional neural network based on Ghost combined with CBAM for extracting image

**Table 4. Comparison of model performance results with other researchers.**

| Method | Accuracy (%) | Model size (MB) | FLOPs |
|--------|--------------|-----------------|--------|
| [27] | 99.76 | 9.25 | 581.71M |
| [28] | 95.00 | 14.3 | 15.80G |
| [29] | 97.95 | 8.58 | 463.28M |
| Ours | 98.58 | 3.61 | 627.69M |

features to identify cows. Using a deep learning convolutional neural network with a large network resnet50 as the skeleton network. First, the large convolution kernel in the first layer is replaced by a small convolution kernel, which is inflated to maintain the perceptual field of the large convolution kernel. Then use plug-and-play ghost module in the model to reduce the model computation, and introduce channel and spatial attention CBAM behind each combined stacked stage, while focusing on more important features from both spatial and channel dimensions to improve the model recognition accuracy. The final model trained and recognized the lateral view images of the collected 13 cows with an average recognition rate of 98.58%, a model size of 3.61MB, and model flops of 827.69M. In comparison with other classical networks, the results show that the model in this study has the lowest number of model parameters while maintaining a high recognition accuracy. Compared with the models proposed by other researchers, the results show that our model has lower model size and FLOPs while maintaining high recognition accuracy. The identification method proposed in this study is a breakthrough and improvement on previous work, and the proposed method can provide a scientific basis for individual identification of dairy cows under practical application, and also lay the theoretical foundation for individual identification of other animals.

## Ethical statement

The cow data used in this study were taken from the local farm, and the farmer's consent has been obtained when taking photos of the cows. The cows used for image shooting are not supervised by the regulatory authorities. No harm has been caused to the cows during the data collection process and the research experiment. The research content does not involve ethics, so ethical approval is not required.

## Author Contributions

**Conceptualization:** Ye Mu.

**Data curation:** Lili Fu, Haohong Pang.

**Formal analysis:** Ruiwen Ni.

**Methodology:** Shuolin Kong.

**Project administration:** Ying Guo.

**Resources:** Tianli Hu.

**Supervision:** He Gong.

**Validation:** Yu Sun.

**Writing – original draft:** Lili Fu.

**Writing – review & editing:** Shijun Li.

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
