## [Decision Letter · Decision Letter 0]

2 Aug 2022

PONE-D-22-15687Lightweight individual cow identification based on Ghost combined with attention mechanismPLOS ONE

Dear Dr. Gong,

Thank you for submitting your manuscript to PLOS ONE. After careful consideration, we feel that it has merit but does not fully meet PLOS ONE’s publication criteria as it currently stands. Therefore, we invite you to submit a revised version of the manuscript that addresses the points raised during the review process.

We look forward to receiving your revised manuscript.

Kind regards,

Tao Song

Academic Editor

PLOS ONE

Journal Requirements:

Reviewers' comments:

Reviewer's Responses to Questions

**Comments to the Author**

1. Is the manuscript technically sound, and do the data support the conclusions?

Reviewer #1: Yes

Reviewer #2: Yes

2. Has the statistical analysis been performed appropriately and rigorously? 

Reviewer #1: Yes

Reviewer #2: Yes

3. Have the authors made all data underlying the findings in their manuscript fully available?

Reviewer #1: Yes

Reviewer #2: Yes

4. Is the manuscript presented in an intelligible fashion and written in standard English?

Reviewer #1: Yes

Reviewer #2: No

5. Review Comments to the Author

Reviewer #1: 1. The data set constructed by itself is not convincing, and the small number of pictures in the data set is even less convincing. The data set should include training data, validation data, and test data. To avoid overlap between training data and test data, training data and test data should come from different data sets.

2. There are many separate drawings and annotations in the paper, for example, Figure 2. Please check carefully and make modifications.

3. When using abbreviations, you should give the full name of the noun before using the abbreviation.

4. It is mentioned in the paper that empty convolution is used to replace ordinary convolution. Please explain the reason for doing so.

5. The ghost module and the convolutional blocking attention module are mentioned to improve ReNet50. Please explain why these two modules are used to improve ReNet50 instead of other modules.

6. The experimental results are not enough to prove that the improved model has a high recognition accuracy. It is hoped that the author can supplement the experiments to prove that the model is more competitive than other models.

7. In terms of identification of cows, heuristic algorithms (https://doi.org/10.1038/s41540-022-00233-w, DOI: 10.1109 / TCBB. 2021.3127271) and self-attention (DOI: 10.1186/ S12864-022-08687-2) and other deep learning models also have great application potential. I hope the author can make a reasonable introduction and analysis.

Reviewer #2: Thanks for your contribution, your work is complete, but some experimental procedures need to be more clearly explained. 1. The explanation of CBAM in the “methodology theory” chapter is redundant with the content in “Convolutional block attention module” chapter. 2. Could you explain stage 1,2,3 in Figure 3, and if they are Ghost Modules, what's the difference between them? 3. Since it is a model built on the basis of ResNet50, why not compare it with ResNet50? 4. The models compared in this paper have been proposed for a long time. Has there been any research on the individual recognition models in recent years?

6. PLOS authors have the option to publish the peer review history of their article (what does this mean?). If published, this will include your full peer review and any attached files.

Reviewer #1: No

Reviewer #2: No

---

## [Author Response · Author response to Decision Letter 0]

10 Aug 2022

Dear Editor and Reviewers，

Thank you for your valuable comments on " Lightweight individual cow identification based on Ghost combined with attention mechanism "(ID: PONE-D-22-15687). Those comments are all valuable and very helpful for revising and improving our paper, as well as the important guiding significance to our researches. We have studied comments carefully and have made correction which we hope meet with approval. The revised part has been used to track revisions and is named "revised manuscript with track changes" to be uploaded as a separate manuscript. The main corrections in the paper and the responds to the reviewer’s comments are as flowing:

Reviewer #1: 

1. Training data and test data overlap

Answer: Thank you very much, I am Sorry for your misunderstanding due to my writing, we have enriched and refined the "data collection and processing" in the manuscript. Our data set is collected from the real farm, and all data has been uploaded to figshare. In this experiment, the data set is divided into 70% of the training set, 20% of the verification set and 10% of the test set. There is no overlapping image between the training data and the test data.

2. There are many separate drawings and annotations in the paper, for example, Figure 2. Please check carefully and make modifications.

Answer: Thanks for the reminder, we have revised it.

3. When using abbreviations, you should give the full name of the noun before using the abbreviation.

Answer: Thanks for the reminder, The full name of the noun has been given before using abbreviations.

4. It is mentioned in the paper that empty convolution is used to replace ordinary convolution. Please explain the reason for doing so.

Answer: Thank you for your comments. In Section 3.1, we refine the reasons for using void convolution instead of ordinary convolution. Ordinary convolution cannot take into account both receptive field and parameter quantity. In order to lighten the model, we choose to use empty convolution that can satisfy large receptive field and less parameter quantity.

5. The ghost module and the convolutional blocking attention module are mentioned to improve ReNet50. Please explain why these two modules are used to improve ReNet50 instead of other modules.

Answer: Thank you for your valuable opinions, firstly, resnet50 is a classical deep convolutional neural network and can be further lightweight. Secondly, as a deep convolution neural network, resnet50 will generate a large number of similar feature maps in the process of calculating features, and ghost module can efficiently generate these similar feature maps through simple linear operation. In addition, the structure of ghost module is very similar to that of resnetblock, so this study uses ghost module to improve resnet50. The addition of attention mechanism is only to further improve the accuracy of the model.

6. The experimental results are not enough to prove that the improved model has a high recognition accuracy. It is hoped that the author can supplement the experiments to prove that the model is more competitive than other models.

Answer: Thank you for your comments, In the manuscript, we compared the model of this study with other classical convolutional neural network models, in which we supplemented the performance comparison of the new model just proposed in the last two years. In addition, we compared the model with the model proposed by other researchers in recent years for individual identification of cows, and compared it with our previous studies. The results showed that our model was improved compared with previous studies, and its performance was better than that of other researchers.

7. In terms of identification of cows, heuristic algorithms (https://doi.org/10.1038/s41540-022-00233-w, DOI: 10.1109 / TCBB. 2021.3127271) and self-attention (DOI: 10.1186/ S12864-022-08687-2) and other deep learning models also have great application potential. I hope the author can make a reasonable introduction and analysis.

Answer: Thank you very much for your comments. We carefully read the two articles recommended by you and cited self-attention (DOI: 10.1186/s12864-022-08687-2), which enriched our manuscript.

Special thanks to you for your good comments. 

Reviewer #2:

1. The explanation of CBAM in the “methodology theory” chapter is redundant with the content in “Convolutional block attention module” chapter.

Answer: Thanks to your valuable comments, we found that it is indeed a duplicate, so we have deleted it, enriched and refined the " Convolutional block attention module"

2. Could you explain stage 1,2,3 in Figure 3, and if they are Ghost Modules, what's the difference between them?

Answer: Thank you for your suggestion, fig. 3 is the model we built, in which stage1, stage2 and stage3 are added with different numbers of ghost in a progressive manner to ensure the depth of the model. At the same time, it follows the rule that the more the feature maps in the forward propagation process of the convolution neural network, the more convolution kernels are required.

3. Since it is a model built on the basis of ResNet50, why not compare it with ResNet50?

Answer: Thank you for your comments, the comparison with resnet50 is not included in the " Model performance validation". In order to better show the effectiveness of the methods we used to improve, we put the comparison results between the model and the original model resnet50 in the " Ablation experiments".

4. The models compared in this paper have been proposed for a long time. Has there been any research on the individual recognition models in recent years?

Answer: Thank you for your comments, in " Model performance validation ", the models we compare are all classical convolutional neural network models. According to your opinion, we have added two new models, EfficientNet v2 (2021) and PVT (2021). In addition, we compared with other methods of studying cows in recent years, and the results are shown in the manuscript.

Special thanks to you for your good comments.

We tried our best to improve the manuscript and made some changes in the manuscript. These changes will not influence the content and framework of the paper. 

We appreciate for Editors/Reviewers’ warm work earnestly, and hope that the correction will meet with approval. Once again, thank you very much for your comments and suggestions.

---

## [Decision Letter · Decision Letter 1]

19 Sep 2022

Lightweight individual cow identification based on Ghost combined with attention mechanism

PONE-D-22-15687R1

Dear Dr. Gong,

We’re pleased to inform you that your manuscript has been judged scientifically suitable for publication and will be formally accepted for publication once it meets all outstanding technical requirements.

Kind regards,

Tao Song

Academic Editor

PLOS ONE

Additional Editor Comments (optional):

Reviewers' comments:

Reviewer's Responses to Questions

**Comments to the Author**

1. If the authors have adequately addressed your comments raised in a previous round of review and you feel that this manuscript is now acceptable for publication, you may indicate that here to bypass the “Comments to the Author” section, enter your conflict of interest statement in the “Confidential to Editor” section, and submit your "Accept" recommendation.

Reviewer #1: All comments have been addressed

Reviewer #2: All comments have been addressed

2. Is the manuscript technically sound, and do the data support the conclusions?

Reviewer #1: Yes

Reviewer #2: Yes

3. Has the statistical analysis been performed appropriately and rigorously? 

Reviewer #1: Yes

Reviewer #2: Yes

4. Have the authors made all data underlying the findings in their manuscript fully available?

Reviewer #1: Yes

Reviewer #2: Yes

5. Is the manuscript presented in an intelligible fashion and written in standard English?

Reviewer #1: Yes

Reviewer #2: Yes

6. Review Comments to the Author

Reviewer #1: The use of accurate and informative dairy farming is very interesting, and the authors have addressed all the comments in this revised version.

Reviewer #2: (No Response)

7. PLOS authors have the option to publish the peer review history of their article (what does this mean?). If published, this will include your full peer review and any attached files.

Reviewer #1: No

Reviewer #2: No

---

## [Editor Report · Acceptance letter]

20 Sep 2022

PONE-D-22-15687R1 

Lightweight individual cow identification based on Ghost combined with attention mechanism 

Dear Dr. Gong:

I'm pleased to inform you that your manuscript has been deemed suitable for publication in PLOS ONE. Congratulations! Your manuscript is now with our production department. 

Kind regards, 

on behalf of

Dr. Tao Song 

Academic Editor

PLOS ONE